# Effects of Diabetes Quality Assessment on Diabetes Management Behaviors Based on a Nationwide Survey

**DOI:** 10.3390/ijerph192315781

**Published:** 2022-11-27

**Authors:** Chang Kyun Choi, Jungho Yang, Ji-An Jeong, Min-Ho Shin

**Affiliations:** 1Institute for Biomedical Science, Chonnam National University Hwasun Hospital, Jeollanam-do 58128, Republic of Korea; 2Department of Preventive Medicine, Chonnam National University Medical School, 264 Seoyang-ro, Jeollanam-do 58128, Republic of Korea

**Keywords:** diabetes complication, health survey, cross-sectional studies, health care quality assessment

## Abstract

Improved diabetes management in primary care is essential for reducing the public health burden of diabetes, and various programs are being implemented in Korea for this purpose. Although the Health Insurance Review and Assessment (HIRA) evaluates the quality of type 2 diabetes management in primary care clinics and hospitals, it is unclear whether the implementation of these evaluations is related to the adequate management of complications in diabetic patients. We evaluated the association between the proportion of clinics managing diabetes well and lifestyles and uptake of screening for complications in 24,620 diabetic participants of the 2019 Korean Community Health Survey (KCHS). Multivariate multilevel logistic regression was performed to evaluate the fixed effect of the district-level variable and the heterogeneity among districts. The proportion of clinics with good diabetes management per 10,000 inhabitants was positively related to screening for diabetes complications. Furthermore, this district variable was significantly related to engaging in walking activity (Odds ratio: 1.39, 95% CI: 1.10–1.76) and sufficiently explained the heterogeneity among districts. However, current smoking and weight control were not associated with the proportion of clinics with good diabetes management. The financial incentives to primary care clinics would improve the primary prevention of diabetic complications.

## 1. Introduction

Globally, diabetes mellitus has caused among the largest increases in the attributable number of disability-adjusted life years (DALYs) over the past 20 years [1]. In South Korea, diabetes is the sixth leading cause of death and is the second-largest disease attribute for DALYs [2]. With the increasing prevalence of diabetes [3], its public health burden is expected to increase. Because these trends are more rapid in Korea than globally [1], interventions are needed to reduce the public health burden of diabetes.

The management of diabetes in South Korea is often ineffective despite good access to care. In Korea, the mean number of physician visits per year is 16.8 for the general population, the highest in Organization for Economic Cooperation and Development (OECD) countries [4], but it is three times higher in diabetic patients [5], and the hospital admission rate for diabetics in South Korea is twice as high as the OECD average [4]. One of the main reasons for the poor outcome of diabetes management is insufficient diabetes management in primary care clinics [6]. Of diabetic patients treated at tertiary hospitals, 91.0% undergo screening tests for diabetic neuropathy within 2 years and 61.5% do so for retinopathy, compared to 32.2% and 31.1%, respectively, among diabetic patients treated at primary care clinics. In addition, there is large among-district disparity for diabetes complication management. According to National Health Insurance Service (NHIS) data [7], fundus examination and microalbuminuria testing varies the most, showing tenfold (0.9–9.2%) and fourfold (6.3–28.9%) variation by district, respectively.

To improve diabetes complication management, various projects have been implemented in South Korea, including the Korean community-based Registration and Management Program for Hypertension and Diabetes [7], the Non-communicable Diseases Prevention and Control Program [8], and the Primary Care Project [9]. Health Insurance Review and Assessment (HIRA) evaluates treatment outcomes in clinics and hospitals for chronic diseases such as hypertension, diabetes, and asthma, and provides financial incentives to primary care clinics and hospitals with good outcomes. The diabetes quality assessment (DQA) includes indices such as continuity of treatment and the proportion of patients who undergo testing for complications. Since its implementation in 2009, the number of primary care clinics with good diabetes outcomes has been increasing steadily. However, no study has of yet evaluated whether there is a relationship between the clinics that manage diabetes well and the quality of diabetes management.

Therefore, this study evaluated the effectiveness of the HIRA DQA by evaluating the association between the number of clinics with good diabetes management and indices related to diabetes management using a nationwide health survey.

## 2. Methods

### 2.1. Study Population

This study included participants with a self-reported diagnosis of diabetes (12,966 men, 13,536 women) in the 2019 Korean Community Health Survey (KCHS), a nationwide, cross-sectional survey conducted by the Korea Disease Control and Prevention Agency (KDCA) [10]. The purpose of KHCS is to produce statistics on health behaviors, compare health indicators among regions, and create evidence for conducting regional health projects. Items on the KCHS include lifestyle, physical activity, oral hygiene, mental hygiene, medical use, quality of life, etc. The KCHS targets adults aged ≥19 years who live within the jurisdiction of a community health center. The stratum was subdivided according to administrative (dong, eup, and myeon) and housing (apartments and houses) units; the smallest administrative district units (tong, ban, and ri) were selected as the primary sampling unit of the stratum through probability proportionate sampling. Sample households were extracted through systemic sampling of the primary sampling units. Information was gathered through face-to-face interviews with all members of the sampled household conducted by trained interviewers. The final analysis included 12,161 men and 12,459 women with no missing values. The KCHS protocol is annually reviewed and approved by the institutional review board of the KDCA. Written informed consent was obtained from all participants in the KCHS. No additional institutional review board approval was required for this study because KCHS is excluded from the review list pursuant to Article 2.2 of Enforcement Rule of Bioethics and Safety Act in Republic of Korea.

### 2.2. Definition of Variables

Smoking status was categorized as never, ex-, and current smoker. Never smokers were defined as participants who did had not smoked more than five packs of cigarettes in their lifetime. Among the participants who smoked more than five packs of cigarettes during their lifetime, participants who did not currently smoke were categorized as ex-smokers, and those who currently smoked were classified as current smokers. Walking activity was dichotomized according to whether participants walked ≥30 min at least 5 days per week. Alcohol consumption was coded as lifetime abstainer, ex-drinker, occasional drinker, and current drinker. Lifetime abstainers were people who had never consumed alcoholic beverages, except during rituals. Occasional drinkers were participants who drink alcoholic beverages less than once a month. Self-rated health (SRH) was measured using a five-point Likert scale (1 = very good to 5 = very poor). Body mass index (BMI) was calculated in kilograms per meter squared from self-reported weight and height. Participants were categorized into five groups according to the WHO Asian classification [11] based on their BMI as underweight (<18.5 kg/m^2^), normal-weight (18.5–22.9 kg/m^2^), overweight (23.0–24.9 kg/m^2^), obese (25.0–29.9 kg/m^2^), and severely obese (≥30.0 kg/m^2^). Household income, marital status, and educational attainment were investigated as indices of socioeconomic status (SES). Equivalized income was calculated by dividing household income by the square root of household size [12] and subdivided by quintile. The cutoff of self-reported household income quintiles using sampling weight was calculated for all participants in the 2019 KCHS. Marital status was based on whether participants were living with a partner. Years of education was categorized as ≤6, 7–8, 9–12, and ≥13 years.

### 2.3. Diabetes Quality Assessment

Since 2011, HIRA has annually evaluated whether diabetic patients are properly managed to reduce the risk of cardiovascular complications with the adequate treatment of diabetes and provided incentives for medical institutions with adequate treatment. DQA has been implemented in all medical institutions that submit medical expense claims for diabetes as either a main or secondary disease and is administered during annual outpatient treatment visits. Medical bills claimed by healthcare providers, including diabetes as a disease and the prescription of hypoglycemic agents, are used as the assessment data. Since 2019, HIRA has published primary care clinics that provide good treatment on its website, along with their addresses. The following indicators are evaluated: as indicators of the continuity of treatment, the proportion of patients who visited a center more than once per quarter and the proportion of prescription days; as prescription indicators, the proportions of duplicated prescriptions involving the same ingredient groups and prescriptions with over four ingredient groups; and the proportion of patients undergoing HbA1c and lipid tests and fundus examination. DQA targets medical institutions with a total number of prescriptions for diabetes medication of 30 or more per year. In 2019, 14,447 medical institutions were evaluated, of which 3920 (27.1%) were evaluated as being adequate. The number of clinics with good diabetes management in 250 districts (Si, Gun, and Gu) in Korea was calculated from the list of primary care clinics with adequate treatment that was published by HIRA. The mid-year population in 2019 for districts was based on data published by the National Statistical Office of Korea. In the analyses, the proportion of primary care clinics with adequate treatment of diabetes per 100,000 inhabitants was used as the district-level variable.

### 2.4. Statistical Analysis

The weighted baseline characteristics of the study population are expressed as the mean ± standard deviation or proportion (%). We used multivariate multilevel logistic regression to evaluate the relationship between individuals and district-level variables and those undergoing screening tests for diabetes complications. For the assessment of lifestyles, smoking history and BMI were dichotomized as current smoker and nonsmoker, and obese and non-obese, respectively. Because the prevalence of smoking was low in women, only men were included in the model for current smoking. Model 0 was the null model, with a random variance component for the 250 administrative districts. Model 1 also adjusted for age and sex. Model 2 further adjusted for lifestyle (smoking, drinking, and walking activity) and health status (SRH and categorized BMI). Model 3 added household income, years of education, and marital status as indices of SES. Finally, Model 4 also adjusted for the proportion of clinics with good diabetes management as the district-level variable. The odds ratio (OR) and its 95% confidence interval (CI) were estimated by multivariate multilevel logistic regression using the Laplace approximation [13].

The intraclass correlation coefficient (ICC) and median odds ratio (MOR) were calculated to assess the importance of district-level effects on lifestyles and screening for diabetic complications [14]. ICC is the proportion of total variance explained at the district level. The MOR converts the district-level variance into an odds ratio scale, which then can be compared directly with the ORs of individual- and district-level variables. The MOR was defined as the median value of the OR between the area at higher and lower propensities of outcome when selecting two areas. If the MOR equals 1, there are no differences among districts in the probability of taking diabetes complication tests.

All results were weighted using the survey weights provided by KCHS. Significance was set at a *p*-value < 0.05. All analyses were performed with R ver. 4.1.0 (R Foundation for Statistical Computing, Vienna, Austria).

## 3. Results

Table 1 shows the weighted baseline characteristics of diabetic patients in 2019 KCHS according to sex. Compared to women, men were younger, more likely to be smokers and drinkers, engaged in more walking activity, had a good SRH and a higher household income and education level, and were more likely to live with a partner and undergo an annual fundus examination and HbA1c test twice a year.

Table 2 shows the relationship between the individual and district-level variables and lifestyles in patients with diabetes. Because the prevalence of smoking was low in women, the logistic regression model for current smoking was restricted to men. Current smokers were younger, less likely to engage in walking activity, had poorer SRH, and had a lower BMI than non-smokers. A high prevalence of obesity was associated with younger age, female sex, lower prevalence of current smoking, less walking activity, and lower educational attainment. Diabetic patients who engaged in walking were more likely to be male, current smokers, have good SRH, higher educational attainment, and live with a partner. The proportion of clinics with good diabetes management was related to the prevalence of walking activity (OR [Odds ratio]: 1.39, 95% CI [Confidence Interval]: 1.10–1.76), but not current smoking (OR: 0.91, 95% CI: 0.81–1.01) or obesity (OR: 0.97, 95% CI: 0.91–1.04).

Table 3 shows the relationships between individual- and district-level variables and the uptake of the complication screening tests. Individual variables associated with annual fundus examinations were older age, female sex, lower prevalence of current smoking, poorer SRH, and higher SES. Undergoing a microalbuminuria test was associated with female sex, lower prevalence of current drinking, and higher SES, and undergoing a HbA1c test was associated with younger age, female sex, lower prevalence of current smoking, and higher SES. The proportion of clinics with good diabetes management was positively associated with undergoing diabetic complication testing. For each increment in the number of clinics with good diabetes management per 10,000 inhabitants, the ORs of undergoing a fundus examination, microalbuminuria test, and HbA1c test were 1.32 (95% CI 1.04–1.66), 1.46 (95% CI 1.14–1.87), and 1.38 (95% CI 1.09–1.76), respectively.

Table 4 shows the random effects for individual- and district-level factors associated with taking diabetes complication tests and lifestyles in diabetes patients. In models for current smoking, the ICC did not differ between the null model and model 3, where all individual variables were adjusted for. Moreover, the proportion of clinics with good diabetes management was not related to the prevalence of current smoking and did not explain the heterogeneity among districts. The heterogeneity of obesity among districts was partially explained by individual variables. Compared with the ICC in the null model, the ICC in model 4 decreased from 0.010 to 0.080. However, the ICC did not decrease further in model 4, which adjusted for the proportion of clinics with good diabetes management. Conversely, the heterogeneity of walking activity was hardly explained by individual variables, instead the proportion of clinics with good diabetes management was fully explained. After additional adjustment for the proportion of primary clinics with good diabetes management, the ICC was less than 0.001. Among individual variables, the heterogeneity among districts was mostly explained by socioeconomic variables. After adding SES variables, the ICC of undergoing a fundus examination, microalbuminuria test, and HbA1c test was reduced from 0.041 to 0.037, 0.065 to 0.062, and 0.062 to 0.057, respectively. However, the proportion of clinics with good diabetes management explained the heterogeneity among districts more than did individual variables. After adding the proportion of clinics with good diabetes management as the district-level variable, the ICC decreased to 0.008, 0.031, and 0.044 in model 4 for undergoing a fundus examination, microalbuminuria test, and HbA1c test, respectively. ORs of the variables according to the models are presented in the Appendix A (Appendix A).

## 4. Discussion

In our study, the proportion of clinics with good diabetes management was positively related to walking activity in patients with diabetes and accounted for the variance among districts. Moreover, the number of clinics with good diabetes management was related to undergoing adequate screening for diabetes complications, and this propensity was similar for the microalbuminuria test, which is not included in DQA. About half of the heterogeneity among districts associated with uptake of the microalbuminuria test was explained by the district-level variable, and some unexplained heterogeneity was explained by the SES of individuals.

The positive relationship between the proportion of hospitals with good diabetes management and walking activity can be explained by the health education provided by medical personnel to diabetic patients to have a good lifestyle. Several large-scale chronic disease management programs implemented in the South Korea Program [8,9] reported that education and consulting improved disease management. However, since the proportion of hospitals with good diabetes management in our study did not show an association between smoking rate and obesity rate, it is possible that the proportion of hospitals with good diabetes management may proxy other characteristics at the regional level.

Because the DQA does not include lifestyle modification indices, follow-up studies on the effects of physical activity on diabetes outcomes are needed. Although several observational studies have found a relationship between walking and mortality in patients with diabetes [15,16,17], a randomized controlled trial (RCT) found no association between physical activity promotion and lower mortality [18]. The discrepancy between the observational and RCT findings may be due to insufficient long-term follow-up of walking interventions. Two RCTs found that while glycemic and weight control improved during a short intervention, the difference between the intervention group and the control group for the corresponding intermediate phenotypes decreased in the follow-up period after the intervention was terminated [18,19], While individual physical activity may be difficult to track over the long-term, the recent widespread use of smartphones and wearable devices has made it possible to track daily steps in real-time [20]. Furthermore, an RCT study found that a digital health intervention helped participants increase their daily steps [21]. Therefore, future NHIS funding of digital therapy for lifestyle modification would enable the DQA to assess lifestyle issues.

We found no significant association between the proportion of clinics with good diabetes management and the prevalence of smoking and obesity. This finding may be explained by the fact that lifestyle modifications for smoking cessation and weight control may have different effects on patients with diabetes and the general population. Nicotine replacement therapy, which is commonly used for smoking cessation, has been shown to increase insulin resistance [22]. Furthermore, diabetes medication often causes weight gain, making weight control more difficult for diabetic patients than for the general population. However, the side effects of medication supporting lifestyle changes have not been sufficiently examined in patients with diabetes [23]. Although the DQA assesses only medication adherence and screening for complications, recent NHIS support for nicotine dependence treatment means that NHIS claim data can be used to assess the effect of smoking cessation programs. Alternatively, our finding that smoking and obesity prevalence were not associated with the number of clinics with good diabetes management may be explained by our methodology. We assessed current smoking and BMI status rather than the effects of lifestyle modification programs because KCHS data do not specify the times of smoking cessation, diabetes diagnosis, and weight changes.

In a previous study that used 2013 KCHS data [24], the number of internists per 1000 inhabitants was positively related to undergoing a fundus examination or microalbuminuria test (OR 1.26, 95% CI 1.01–1.57), and this effect size was smaller than that in our study. The heterogeneity among districts explained by the district-level variable was greater in our study than in the previous study. In our study, after adding the district-level variable to the model, ICC decreased by 3.2% and 2.8% in the models for undergoing microalbuminuria and fundus examinations, respectively. In the previous study, ICC decreased by 1% after adding district-level variables to the model. This discrepancy is due to differences in the characteristics of the district-level variables. The proportion of internists simply reflects the distribution of internists regardless of the type of medical institute with which the internist is affiliated, whereas the proportion of clinics with good diabetes management reflects both quantitative and qualitative aspects of primary care clinics that care for diabetic patients. In 2019, there were 32,169 primary clinics in South Korea, of which 14,447 (45.0%) issued more than 30 prescriptions for diabetes medication annually, and those clinics were assessed using the DQA. Therefore, the effect size of the proportion of clinics with good diabetes management was greater than that of the proportion of internists.

The heterogeneity among districts explained by the proportion of primary clinics with good diabetes management differed according to the uptake of diabetic complication tests, which may be due to differences in treatment adherence in the various regions. Unlike the microalbuminuria and HbA1c tests, which are performed in a laboratory, fundus examinations must be performed by an ophthalmologist. A previous study that used NHIS data to assess diabetes management in various districts found that the relative variance in uptake of fundus examinations was greater than that for the microalbuminuria and HbA1c tests [7]. Moreover, the quality of diabetes care is higher in urban areas than in suburban or rural areas. The KCHS was designed to assess general health behaviors and health services in administrative districts; therefore, our analysis, which was restricted to diabetic patients, did not generate meaningful statistics for administrative districts.

We found that higher SES was associated with good management of diabetic complications. According to the 2015 KCHS, diabetic patients with lower educational attainment or lower household incomes were less likely to undergo annual fundus examinations or microalbuminuria tests [25]. Several previous studies have shown an association between SES and the risk of diabetes complications. Data from the Korean National Diabetes Program [26] and the Korea National Health and Nutrition Examination Survey V [27] revealed that diabetic patients with higher SES were more likely to have controlled hyperglycemia and were at lower risk of diabetic retinopathy. Similarly, in Singapore, lower SES was associated with a higher incidence and more rapid progression of diabetic retinopathy [28]. Furthermore, the risk of acute kidney injury and end-stage renal disease is higher in diabetic patients with lower SES [29,30]. Therefore, a public health policy to increase the uptake of screening for diabetic complications is needed to manage patients with lower SES.

Our study had several limitations. First, the monofilament test, which is a screening tool for diabetic neuropathy, was not included in the analysis. Instead, we used the microalbuminuria test, which is not included in the DQA; we found a positive association between clinics with good diabetes management and annual microalbuminuria tests. Second, further studies are needed to compare health outcomes (e.g., mortality from diabetic complications) in patients treated at adequate primary care clinics and untreated patients. Third, the confounding effects of district-level variables, such as the distribution of commercial laboratories and ophthalmology clinics, need to be taken into consideration. Fourth, because KCHS is a face-to-face survey based on self-reported data, the rate of testing for complications may be over-reported. For example, the 2015 KCHS reported that 38.15% of diabetic patients underwent microalbuminuria tests and 33.9% underwent fundus examinations [25], whereas the respective percentages were 15.4% and 2.9% according to NHIS claims data [7]. Fifth, we did not evaluate the effect of DQA on treatment modalities such as usage of an insulin pump or the number of oral hypoglycemic agents being taken because that information was not included in KCHS. Sixth, because the KCHS is a questionnaire-based survey, recall bias cannot be ruled out in our study.

## 5. Conclusions

The proportion of adequate primary care clinics was positively associated with walking activity and screening for diabetic complications. Future studies should include various district-level variables that assess walkability and medical adherence.

## Figures and Tables

**Table 1 ijerph-19-15781-t001:** Weighted baseline characteristics of diabetes patients in 2019 KCHS according to sex.

	Men	Women	*p* Value
Unweighted N.	12,161	12,459	
Age (years)	61.8 ± 12.5	66.0 ± 12.1	<0.001
Smoking history			<0.001
Never	19.1	93.4	
Ex-smoker	49.9	3.2	
Current smoker	31.0	3.5	
Alcohol consumption			<0.001
Lifetime abstainer	8.4	36.2	
Ex-drinker	23.2	23.9	
Occasional drinker	10.1	19.8	
Current drinker	58.3	20.2	
Walking activity ^a^	47.2	42.4	<0.001
Self-rated health	3.2 ± 0.9	3.5 ± 0.8	<0.001
Body mass index (kg/m^2^)			0.440
<18.5	1.7	2.1	
18.5–22.9	24.7	27.5	
23.0–24.9	27.0	22.3	
25.0–29.9	39.3	38.1	
≥30.0	7.2	10.1	
Equivalized household income			<0.001
First quintile	31.8	44.7	
Second quintile	20.4	20.1	
Third quintile	14.9	13.5	
Fourth quintile	16.3	12.1	
Fifth quintile	16.5	9.6	
Years of education			<0.001
≤6	17.4	49.1	
7–9	15.9	16.5	
10–12	37.2	25.1	
≥13	29.5	9.3	
Living with a partner	81.4	59.6	<0.001
Taking fundus examination annually	44.7	47.5	<0.001
Taking microalbuminuria test annually	54.6	54.4	0.777
Taking HbA1c test at least twice a year	59.1	54.9	<0.001

All values were calculated by using sampling weights; Values were presented as percentage or ‘mean ± standard deviation’; ^a^ Walking for more than or equal to 30 min on at least 5 days per week; KCHS: Korean Community Health Survey; HbA1c: glycated hemoglobin.

**Table 2 ijerph-19-15781-t002:** Multilevel multivariate logistic regression for lifestyles in diabetes patients.

	Current Smoking ^a^	Walking Activity ^b^	Obesity
Districts level			
N. of primary clinics with good adequacy per 10,000 inhabitants	1.10 (0.89–1.36)	1.39 (1.10–1.76)	0.90 (0.78–1.04)
Individuals level			
Age (10 years)	0.95 (0.95–0.95)	0.98 (0.96–1.01)	0.74 (0.72–0.76)
Women	-	0.90 (0.82–0.98)	1.11 (1.02–1.21)
Smoking history			
Never-smoker	-	1 (reference)	1 (reference)
Ever-smoker	-	1.05 (0.96–1.14)	1.16 (1.07–1.27)
Current smoker	-	0.84 (0.76–0.92)	0.75 (0.69–0.83)
Drinking behavior			
Lifetime abstainers	1 (reference)	1 (reference)	1 (reference)
Ex-drinkers	0.88 (0.73–1.07)	0.92 (0.84–1.01)	0.95 (0.87–1.04)
Occasional drinkers	1.32 (1.07–1.62)	1.04 (0.94–1.15)	0.97 (0.88–1.07)
Current drinkers	1.98 (1.67–2.35)	0.94 (0.86–1.03)	1.05 (0.96–1.14)
Walking activity ^b^	0.86 (0.79–0.93)	-	0.84 (0.79–0.89)
Poor self-rated health	1.17 (1.11–1.24)	0.75 (0.72–0.78)	0.99 (0.96–1.03)
Body mass index (kg/m^2^)			
<18.5	1.28 (0.93–1.74)	0.66 (0.53–0.81)	-
18.5–22.9	1 (reference)	1 (reference)	-
23.0–24.9	0.66 (0.59–0.74)	0.87 (0.80–0.94)	-
25.0–29.9	0.56 (0.50–0.62)	0.80 (0.75–0.86)	-
≥30.0	0.41 (0.34–0.48)	0.64 (0.57–0.72)	-
Equivalized incomes			
First quintile	1 (reference)	1 (reference)	1 (reference)
Second quintile	0.94 (0.83–1.06)	1.01 (0.93–1.10)	1.00 (0.92–1.08)
Third quintile	0.95 (0.82–1.09)	1.14 (1.04–1.24)	0.96 (0.88–1.05)
Fourth quintile	1.02 (0.89–1.17)	0.96 (0.87–1.05)	1.08 (0.99–1.18)
Fifth quintile	0.85 (0.74–0.98)	0.98 (0.89–1.08)	0.96 (0.87–1.05)
Years of education			
≤6	1 (reference)	1 (reference)	1 (reference)
7–9	0.82 (0.70–0.96)	1.12 (1.02–1.22)	0.79 (0.73–0.87)
10–12	0.91 (0.79–1.04)	0.95 (0.88–1.04)	0.71 (0.66–0.77)
≥13	0.61 (0.53–0.71)	1.14 (1.03–1.26)	0.76 (0.69–0.84)
Living with a partner	0.91 (0.81–1.01)	1.07 (1.00–1.14)	0.97 (0.91–1.04)

All values were presented as “odds ratio (95% confidence interval)”; ^a^ Because of low prevalence of smoking, women were excluded in model; ^b^ Walking for more than or equal to 30 min on at least 5 days per week; HbA1c: glycated hemoglobin; ICC: intra-class coefficient correlation; MOR: median odds ratio.

**Table 3 ijerph-19-15781-t003:** Multilevel multivariate logistic regression for taking diabetes complication test in diabetes patients.

	Fundus Examination Once a Year	Microalbuminuria Test Once a Year	HbA1c Twice a Year
Districts level			
N. of primary clinics with good adequacy per 10,000 inhabitants	1.32 (1.04–1.66)	1.46 (1.14–1.87)	1.38 (1.09–1.76)
Individuals level			
Age (10 years)	1.09 (1.06–1.12)	1.00 (0.97–1.03)	0.86 (0.83–0.89)
Women	1.17 (1.07–1.28)	1.12 (1.02–1.22)	1.27 (1.16–1.40)
Smoking history			
Never-smoker	1 (reference)	1 (reference)	1 (reference)
Ever-smoker	1.00 (0.91–1.09)	1.09 (1.00–1.19)	1.25 (1.13–1.37)
Current smoker	0.76 (0.69–0.84)	0.91 (0.83–1.01)	0.89 (0.80–0.98)
Drinking behavior			
Lifetime abstainers	1 (reference)	1 (reference)	1 (reference)
Ex-drinkers	1.01 (0.93–1.11)	1.01 (0.93–1.11)	0.97 (0.89–1.07)
Occasional drinkers	1.08 (0.98–1.19)	0.96 (0.87–1.06)	1.09 (0.98–1.21)
Current drinkers	0.92 (0.84–1.00)	0.86 (0.78–0.94)	0.93 (0.84–1.01)
Walking activity ^a^	1.20 (1.13–1.27)	1.10 (1.04–1.16)	1.11 (1.05–1.18)
Poor self-rated health	1.21 (1.17–1.25)	1.15 (1.11–1.19)	1.11 (1.07–1.15)
Body mass index (kg/m^2^)			
<18.5	0.90 (0.73–1.11)	0.78 (0.63–0.96)	1.05 (0.85–1.30)
18.5–22.9	1 (reference)	1 (reference)	1 (reference)
23.0–24.9	1.09 (1.00–1.17)	1.09 (1.01–1.18)	1.08 (0.99–1.17)
25.0–29.9	0.96 (0.89–1.03)	0.98 (0.91–1.05)	1.00 (0.93–1.08)
≥30.0	0.93 (0.83–1.04)	0.98 (0.88–1.10)	0.98 (0.87–1.10)
Equivalized incomes			
First quintile	1 (reference)	1 (reference)	1 (reference)
Second quintile	1.07 (0.99–1.16)	1.06 (0.98–1.15)	1.15 (1.06–1.24)
Third quintile	1.00 (0.91–1.10)	0.97 (0.88–1.06)	1.44 (1.31–1.58)
Fourth quintile	1.11 (1.01–1.22)	1.04 (0.95–1.15)	1.21 (1.10–1.33)
Fifth quintile	1.33 (1.21–1.47)	1.32 (1.20–1.46)	1.44 (1.30–1.59)
Years of education			
≤6	1 (reference)	1 (reference)	1 (reference)
7–9	1.23 (1.13–1.35)	1.13 (1.04–1.24)	1.40 (1.28–1.53)
10–12	1.49 (1.37–1.62)	1.35 (1.24–1.47)	1.90 (1.74–2.07)
≥13	1.60 (1.45–1.77)	1.38 (1.25–1.53)	2.28 (2.05–2.53)
Living with a partner	1.14 (1.07–1.22)	1.20 (1.12–1.28)	1.27 (1.19–1.36)

All values were presented as “odds ratio (95% confidence interval)”; ^a^ Walking for more than or equal to 30 min on at least 5 days per week; HbA1c: glycated hemoglobin; ICC: intra-class coefficient correlation; MOR: median odds ratio.

**Table 4 ijerph-19-15781-t004:** Random effect analysis for individual- and district-level factors associated with taking diabetes complication screening tests and lifestyles in diabetes patients.

	Model 0	Model 1 ^a^	Model 2 ^b^	Model 3 ^c^	Model 4 ^d^
Current smoking	0.020	0.021	0.020	0.020	0.041
Walking activity	0.038	0.038	0.038	0.038	< 0.001
Obesity	0.010	0.009	0.008	0.008	0.008
Fundus examination	0.041	0.041	0.041	0.037	0.008
Microalbuminuria test	0.065	0.064	0.065	0.062	0.031
HbA1c test	0.062	0.067	0.066	0.057	0.044

All values were presented as intraclass correlation coefficient; ^a^ Age and sex were adjusted; ^b^ Smoking, alcohol consumption, and body mass index were additionally adjusted; ^c^ Household income, education attainment, and marital status were additionally adjusted; ^d^ The number of primary clinics with good adequacy per 10,000 inhabitants was additionally adjusted.

## Data Availability

Data from the Korea Community Health Survey is available in the Korea Disease Control and Prevention Agency website (https://chs.kdca.go.kr/chs/), accessed on 14 October 2020.

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
