# Peer review of "Effects of Diabetes Quality Assessment on Diabetes Management Behaviors Based on a Nationwide Survey"

_ijerph, 2022, doi:10.3390/ijerph192315781_

Round 1
Reviewer 1 Report
Overall, the article was interesting. I have some concerns listed as below.
Line 80. Like alcohol consumption, authors should give a definition of never, ex, and current smoker. The definition of walking activity should also be described in methods.
Line 85. Which scale was self-rated health measured by?
Line 147. The authors should explain why they divided BMI group into 18.5-22.9 and 23.0-24.9 as they were both considered normal.
As shown in Table 1, there were differences in several characteristics at baseline. The authors did not explain the reasons behind this, and I suspect it would bias the subsequent statistical analysis.
Line 163. Criteria for clinics with "good adequacy" should be defined.
Line 223. The authors should put more efforts in describing why walkability can explain your results. As stated by the authors, walkability is related to the availability of primary care clinics. However, this does not necessarily mean that these clinics manage diabetes well, which is the object of this study.
Line 275. Is it 30 prescriptions for one patient each year or is it the total number of prescriptions a clinic has in a year? It should be clarified.
Line 304. Selection bias such as recall bias should be discussed.
It seems the major analysis of this study focused only on lifestyle and screening tests, so I suggest adjusting the article title to better summarize the main content.
Author Response
We would like to say thank you to the reviewers for the useful comments to improve the paper. We have addressed all the comments as explained below.
Reviewer 1
Overall, the article was interesting. I have some concerns listed as below.
Line 80. Like alcohol consumption, authors should give a definition of never, ex, and current smoker. The definition of walking activity should also be described in methods.
Response: According to your comment, we described the definition of smoking history (lines 87–91). The definition of walking activity was described in the methods (lines 91–92).
Line 85. Which scale was self-rated health measured by?
Response: Five-point scale was used to measure SRH (lines 96–97).
Line 147. The authors should explain why they divided BMI group into 18.5-22.9 and 23.0-24.9 as they were both considered normal.
Response: We categorize BMI based on Asian classification, which is described in methods (lines 98–102).
As shown in Table 1, there were differences in several characteristics at baseline. The authors did not explain the reasons behind this, and I suspect it would bias the subsequent statistical analysis.
Response: Table 1 presents the characteristics of participants according to sex, and since all variables included in Table 1 were corrected in the multivariate analysis, it is thought that bias due to differences in these variables was excluded.
Line 163. Criteria for clinics with "good adequacy" should be defined.
Response: Unfortunately, HIRA discloses evaluation items, but does not disclose weights for each item or scores of medical institutions. Instead, in the methods, we described the proportion evaluated as good adequacy among medical institutions (lines 124–126). In 2019, 3,920 (27.1%) of 14,447 medical institutions subject to evaluation were evaluated as good adequacy.
Line 223. The authors should put more efforts in describing why walkability can explain your results. As stated by the authors, walkability is related to the availability of primary care clinics. However, this does not necessarily mean that these clinics manage diabetes well, which is the object of this study.
Response: According to your comment, we corrected the discussion (lines 239–246).
Line 275. Is it 30 prescriptions for one patient each year or is it the total number of prescriptions a clinic has in a year? It should be clarified.
Response: According to your comment, we describe the condition of the medical institutions evaluated at DQA in the methods (lines 123–124).
Line 304. Selection bias such as recall bias should be discussed.
Response: According to your comment, we discussed the possibility of recall bias in the limitations (lines 352–353).
It seems the major analysis of this study focused only on lifestyle and screening tests, so I suggest adjusting the article title to better summarize the main content.
Response: According to your comment, we corrected the title (lines 2–3).
Reviewer 2
Abstract:
- The last sentence of the abstract seems out of place to me and not an accurate conclusion from the study.
Response: According to your comment, we corrected the last sentence of abstract (line 22).
Introduction:
- Is this concerning type 1 diabetes, type 2 diabetes, or both?
Response: In KCHS, type 2 diabetes was surveyed. We described the type of diabetes in the manuscript.
Methods:
- In the study population, can you explain more about what type of information was collected from the survey? Just a general overview would be helpful.
Response: According to your comment, we described the purpose and main item of KCHS in the method (lines 70–73).
- In the study population, is everyone in the household interviewed or is just one member per household?
Response: KCHS surveys all members of the sampled household. This is described in the method (line 79).
- In the diabetes quality assessment section, how is a clinic classified as having good diabetes management? In other words, how are those indicators combined to determine if the clinic is good or bad at diabetes management?
Response: Unfortunately, HIRA discloses evaluation items, but does not disclose weights for each item or scores of medical institutions. Instead, in the methods, we described the proportion evaluated as good adequacy among medical institutions (lines 124–126). In 2019, 3,920 (27.1%) of 14,447 medical institutions subject to evaluation were evaluated as good adequacy.
- Is there any information about other comorbidities?
Response: Unfortunately, in 2019 KCHS, the only chronic diseases included in KCHS are hypertension and diabetes.
- In the statistical analysis section, do the sample weights come from the KCHS?
Response: Yes. KCHS provided the survey weight. It is described in the last paragraph of the statistical analysis that the survey weight provided by KCHS was used in the analysis (line 155).
Discussion:
- Is there any information on how individuals manage their diabetes – insulin pumps, medication, etc.? If not, this would be good to include in the discussion section as a potential limitation.
Response: Unfortunately, in 2019 KCHS, only the use of insulin and oral hypoglycemic agents was investigated. According to your comment, in the limitations, we described the that assessment of treatment modality was insufficient in our study (lines 350–352).
Reviewer 2 Report
Thank you for the opportunity to review your manuscript “Effects of Diabetes Quality Assessment on Diabetes Management based on a Nationwide Survey”. The paper was very interesting and provided great insight into the effects of diabetes quality assessment on diabetes management. The sections were well defined and followed a logical progression. Overall, I enjoyed the manuscript and believe that it could be a value to our readers.
Abstract:
· The last sentence of the abstract seems out of place to me and not an accurate conclusion from the study.
Introduction:
· Is this concerning type 1 diabetes, type 2 diabetes, or both?
Methods:
· In the study population, can you explain more about what type of information was collected from the survey? Just a general overview would be helpful.
· In the study population, is everyone in the household interviewed or is just one member per household?
· In the diabetes quality assessment section, how is a clinic classified as having good diabetes management? In other words, how are those indicators combined to determine if the clinic is good or bad at diabetes management?
· Is there any information about other comorbidities?
· In the statistical analysis section, do the sample weights come from the KCHS?
Discussion:
· Is there any information on how individuals manage their diabetes – insulin pumps, medication, etc.? If not, this would be good to include in the discussion section as a potential limitation.
Author Response

(The authors gave the same response as above.)
